# Integrated Genetic and Protein Mechanisms Underlying Glucagon-like Peptide-1 Receptor Agonists in Treating Diabetes Mellitus and Weight Loss

**DOI:** 10.3390/cimb47121007

**Published:** 2025-11-30

**Authors:** Lucas Francis, Merlin G. Butler

**Affiliations:** 1College of Osteopathic Medicine, Kansas City University, Kansas City, MO 64106, USA; lucas.francis@kansascity.edu; 2Departments of Psychiatry & Behavioral Sciences and Pediatrics, University of Kansas Medical Center, Kansas City, KS 66160, USA

**Keywords:** glucagon-like peptide-1 (GLP1) receptor (GLP1R) gene, GLP1R agonist, gene-gene or protein interactions, biological processes, functions, pathways and associated diseases, obesity, weight loss, insulin, glucagon, type 2 diabetes mellitus

## Abstract

Glucagon-like peptide-1 receptor (GLP1R) agonists, such as semaglutide, are used for treating type 2 diabetes mellitus and promoting weight loss. This study investigates genetic and molecular mechanisms underlying GLP1R activation using a novel in silico approach to identify effects on metabolism, glucose and insulin production, gastrointestinal motility, behavior, and satiety. This approach used three separate searchable web-based programs and databases (STRING, Pathway Commons, and BioGRID) to identify and analyze functional gene and protein interactions with mechanisms to query GLP1R and related metabolic and appetite regulatory networks with disease associations. We examined integrated gene–gene and protein–protein interactions, pathways, molecular functions, associated diseases, and biological processes for GLP1R, that reportedly involved in diabetes and obesity. GLP1R signaling cascades were described with the activation of the adenylate cyclase-modulating G protein-coupled receptor and increased intracellular cyclic AMP, collectively impacting glucagon production, insulin, glycogenolysis, vasoactive intestinal peptide, and other peptides and hormones required for satiety. Additional factors found were obesity-related peptides (i.e., POMC), hormone signaling, renin secretion, electrolytes and diuresis, circadian rhythm, and insulin secretion. These associations and interactions shift from hypoglycemia to broader endocrine dysfunction. A relationship was noted for GNAS having a role in growth, electrolytes, and skeletal disturbances with specific hormone sensitivity patterns. Understanding established and new interactions with genetics and gene-protein variants that impact type 2 diabetes and obesity would provide further insight into therapeutic GLP1R agonists response and consequences. Potential long-term systemic effects should be monitored, studied, and recorded with treatment protocols adjusted accordingly.

## 1. Introduction

Diabetes mellitus and obesity are major global health burdens affecting millions worldwide. The prevalence of both diagnosed and undiagnosed diabetes is 14.7% for all U.S. adults and increased with age (www.cdc.gov/diabetes/php/data-research/index.html (accessed on 5 April 2025)). More than two in five U.S. adults have obesity (www.cdc.gov/obesity/adult-obesity-facts/index.html (accessed on 5 April 2025)), which is often found in those with type 2 diabetes. Diabetes mellitus is considered a metabolic disease characterized by hyperglycemia with serious complications of nephropathy, neurological deficits, retinopathy, coronary heart disease, and strokes, all contributing to mortality [1,2]. Type 2 diabetes mellitus is one of three types (type 1, type 2, and pregnancy) accounting for more than 90% of cases, with genetic factors playing a role [1,3,4,5,6,7]. It correlates with weight gain and defective insulin production, secretion, and function [5]. To better understand the causes and mechanisms leading to type 2 diabetes, the identification of pathophysiological findings and the role of genetic–protein interactive mechanisms will be required. Since approval by the FDA, semaglutide, a glucagon-like peptide-1 (GLP1) receptor (GLP1R) agonist, has been prescribed to treat type 2 diabetes, commonly marketed under the brand names Ozempic, Wegovy, and Rybelsus, and weight loss in the past few years [8,9].

GLP1R agonists are analogs of endogenous GLP1, a peptide with an incretin effect that enhances insulin secretion after food intake [10]. GLP1 is secreted by the L-cells in the intestine and plays a role in glycemic regulation and satiety signaling [9,10,11,12]. Semaglutide mimics physiological effects leading to weight loss by acting on appetite control centers in the hypothalamus and brainstem [11,12]. However, GLP1R agonists can cause unexpected nervous system changes, retinopathy, and gallbladder and cardiovascular issues [12], requiring more research.

The binding of agonists to the GLP1 receptor protein encoded by the *GLP1R* gene leads to the inhibition of glucagon secretion, increased insulin release, delayed gastric emptying, and decreased appetite. These effects invoke GLP1 binding to GLP1R by stimulating adenylyl cyclase and G-protein signaling to promote insulin synthesis with release [12]. Normal *GLP1* and *GLP1R* genes and their related biological processes, pathways, and functional mechanisms are apparently required for GLP1R agonist activity to treat patients, a focus of our investigations.

We undertook an in silico analysis of GLP1R using updated integrated web-based programs and curated databases to examine gene–gene and protein–protein interactions affecting the pathways, molecular functions, biological processes, and associated diseases. The web-based integrated genetic and protein mechanistic approaches to study GLP1R and its current status, impacted by GLP1R drugs often prescribed and driven by the growing need to treat type 2 diabetes and obesity. However, overlapping mechanisms which undoubtedly relate to optimal drug action and dosage for success treatment may depend on the genetics status, mechanisms of action, processes, disease, and progression per patient. Hence, understanding the genetic status may have the potential to advance optimal medical care, treatment success, and outcomes of affected patients.

We propose that the functional mechanisms of action might also be altered or influenced by the dosage of GLP1R agonists prescribed and directly impact overall health and treatment. In addition, specific gene variants and heterogeneity could lead to disturbed biology, influencing treatment response. Genetic and/or protein testing and monitoring functional mechanisms for GLP1R agonists may be needed, but most individuals with type 2 diabetes and obesity respond positively to treatment in the short term, there are exceptions.

## 2. Materials and Methods

### 2.1. Searchable Literature and Web-Based Programs and Database

#### 2.1.1. Literature and Websites Queried

A comprehensive search was conducted from the published literature and the use of updated interactive web-based programs and databases by focusing on keywords “glucagon-like peptide-1 and receptor (GLP1R)”, “glucagon”, “protein and gene defects/variants”, and “obesity” using the PUBMED database (www.pubmed.org) (accessed on 1 March 2025), focusing only on humans (*Homo sapiens*). Other searchable sites were used to analyze human disorders, their genes, and protein structures and functions. These include the Online Mendelian Inheritance in Man (OMIM) (www.omim.org), a catalog of human genetic disorders and genes, UniProt (www.uniport.org), a database of protein sequences and functional information with the ability to search, align, map, annotate, and download proteins with entries derived for genomic sequencing projects, Ensembl (www.ensembl.org), a genome browser for vertebrates to support research in comparative genomics, evolution, sequence variation, and transcriptional regulation. Gene Cards (www.genecards.org) and Gene Reviews (www.genereviews.org) were searched for up-to-date gene-related information.

#### 2.1.2. Searchable Web-Based Programs and Databases Queried

##### STRING Web-Based Integrated Programs and Databases

We used STRING web-based integrated programs and databases (https://string-db.org) [13,14] searched in July 2025 with versions 12 and 12.5. This web-based program is established and has been used in at least 120 peer-reviewed publications in the literature (PUBMED). We recorded the predicted protein–protein associations, functional interactions, and biological networks to and with top biological processes, molecular functions, cellular components, pathways, and disease–gene associations when searching GLP1R. This web-based program compiles, stores, and integrates curated updated information from peer-reviewed literature and other sources (e.g., PUBMED, OMIM). The documented experimental assay data used by STRING formulates computational predictions of comprehensive objective protein networks that encompass both functional and physical interactions. It then provides a statistical analysis of genes and their encoded protein dataset, adjusts for false discovery, and generates *p*-values. It further provides a visualization interface for interpreting interactive protein–protein associations and generates figures.

Query results can then be displayed as maps that integrate curated experimental data with computational predictions to illustrate relationships among proteins. This visualization enables the identification of potential regulatory nodes, signaling cascades, and functional clusters relevant to gene expression and disease mechanisms. The identified protein clusters are linked to Gene Ontology for biological processes, molecular functions, and cellular components, as well as KEGG and Reactome pathways. This approach provides an updated systematic means to evaluate protein–protein interactions and generate hypotheses about shared biological roles, molecular pathways, and disease associations.

STRING utilizes four separate criteria, including the ***Count in network*** function. This criterion indicates how many proteins are in the visualized protein network and annotated by a particular term based on how many proteins use a specific assigned term. A second statistical factor is ***Strength***, which describes the size of the enrichment effect as a ratio of the number of annotated termed proteins in a network and visualized by the number of proteins expected to be annotated in a random network of the same size [log10 (observed/expected)]. The third factor for statistical analysis is the ***False discovery rate (FDR)***, which examines the significance of enrichment using a measurement to conceptualize the rate of type I or false positive errors in the null hypothesis testing. The reported *p*-values generated by the program are corrected for multiple tests within each category. The ***Signal*** factor is defined as a weighted harmonic mean between the observed/expected ratio and -log (FDR), meant to balance the metrics of larger and smaller terms for more intuitive ordering of enriched terms.

##### Pathway Commons Web-Based Program for Gene–Gene Interactions

We also utilized Pathway Commons (https://www.pathwaycommons.org/), a separate web-based program to identify gene–gene functional interactions and gene regulatory networks. These integrations can impact binding and co-expression or shared processes needed for comparison with results of other programs, such as the STRING program with protein–protein interactions. This website focuses on biological and molecular mechanisms that enable insight regarding conserved networks and pathways relevant to health and diseases with shared processes or functions.

##### Biological General Repository for Interaction Datasets (BioGRID) Web-Based Program for Protein–Protein Interactions

We utilized the searchable BioGRID web-based program and database (https://thebiogrid.org/) that identifies and validates related functional protein–protein interactions and networks. This updated, curated web-based program allows for the study of multiple organisms, including humans. The study sources are considered relevant and assembled by experts prior to uploading into databases for search and analysis. The descriptive computational web-based studies for genetic and protein factors contributing to type 2 diabetes and obesity were undertaken to enrich our primary goal to expand and update our current functional knowledge on how GLP1R agonists can treat type 2 diabetes, obesity, and appetite control by the identification of mechanisms of action.

## 3. Results

### 3.1. STRING Protein–Protein Interactions, Functions, and Analysis

The described web-based programs and databases were analyzed collectively using in silico approaches for functional and genetic mechanisms of the GLP1R. The STRING computerized program and database were utilized to show the top protein–protein associations for GLP1R. The accompanying Tables and Figures describe and predict GLPIR functions and interactions. Figure 1 illustrates the 10 associated protein–protein interactions and networks for GLP1R.

Table 1 presents the top 10 proteins most significantly associated with the GLP1R protein and those identified with the first-tier STRING analysis program. GLP1R is a receptor for GLP1 that binds and mediates insulin secretion from pancreatic beta cells and undertakes a G-protein-dependent process. Different protein–protein associations were identified to further explain the apparent weight loss seen in GL1PR agonist treatment for type 2 diabetes and obesity. Table 1 includes each protein’s symbol and a brief description of the biological role and function. Ligand binding further triggers activation of a signaling cascade that leads to activation of adenylyl cyclase and increases intracellular cyclic AMP (cAMP) levels that regulate insulin secretion in response to GLP1 controlling glucose and insulin levels for appetite control and glycemic effects.

Table 2 summarizes predicted functions of GLP1R based on gene ontology terms and categorizes the data into biological processes, molecular functions, KEGG and Reactome pathways, and disease–gene associations as components of the STRING database analysis. Together, the information from Table 1 and Table 2 establishes the connection of how GLP1R and interacting proteins can contribute to metabolic and endocrine functions, energy balance, and therapeutic effects by GLP1R agonists, important for the treatment of type 2 diabetes and obesity.

In the second-tier STRING web-based analysis, 20 associated proteins with 147 edges were identified with their predicted functions, including 10 proteins found in Figure 1. The 10 new proteins included secretin (SCT), a hormone involved in regulation of pH from the duodenum; vasoactive intestinal polypeptide receptor 2 (VIPR2) for vasoactive intestinal protein (VIP); gastric inhibitory polypeptide receptor (GIPR) for gastric inhibitory peptide (GIP), and glucose-dependent insulinotropic receptor (GPR119) for endogenous fatty acids; melanocortin-2 receptor (MC2R) for adrenocorticotropic hormone and prostaglandin E2 receptor EP2 subtype (PTGER2) for prostaglandin E2; parathyroid hormone receptor 2 (PTH2) and 5-hydroxytryptamine receptor 6 (HTR6) for 5-hydroxytryptamine (serotonin). The top biological processes are adenylating cyclase activation and the modulation of G protein-coupled signaling pathways, for positive regulation of cAMP-mediated signaling. The top molecular functions were peptide binding, hormone receptor binding, and neuropeptide binding. The top KEGG pathways were the cAMP signaling pathway, neuroactive ligand receptor interaction, and insulin secretion. The top Reactome pathways were glucagon-type ligand receptor, G protein alpha signaling events, and secretin family receptors. The top disease–gene associations were hypoglycemia, hyperglycemia, and primary hyperaldosteronism.

The third-tier STRING analysis found 30 associated proteins with 288 edges, with predicted functions for 20 proteins included in the first and second-tier analyses. The new proteins included dopamine (1B) receptor (DRD5), a receptor mediated by G proteins, thyrotropin receptor (TSHR) for thyroid-stimulating hormone, lutropin-choriogonadotropin hormone receptor (LHCGR), and parathyroid hormone receptor 2 (PTH2R). The pituitary adenylate cyclase-activating polypeptide type 1 receptor (ADCYAP1R) is for the pituitary adenylate cyclase-activating polypeptide function. Calcitonin gene-related peptide 2 (CALCB) induces vasodilation, while the Islet amyloid polypeptide (IAPP) inhibits stimulation from insulin to deposit glycogen and the use of glucose. Somatoliberin (GHRH) is known as a growth-releasing hormone and is released by the hypothalamus. It also stimulates the secretion of growth hormone. Vasoactive intestinal polypeptide receptor 1 (VIPR1) is also a receptor for vasoactive intestinal peptide and glucagon receptor (GCGR), a receptor for glucagon and regulation of glucose levels; both peptides and their function are key to understanding diabetes and obesity.

### 3.2. Pathway Commons Gene-Gene Interactions and Functions

The Pathway Commons program analysis of GLP1R found 24 other related or interactive genes, as shown in Figure 2. The interactions between genes and proteins were categorized into distinct types based on their nature of association. Binding interactions refer to physical associations where two or more molecules directly attach to each other, typically forming a complex that may be essential for signal transduction, structural support, or enzymatic activity. Modification interactions represent biochemical changes to a protein that can alter the protein’s function. Post-translational modifications are crucial in regulating signaling pathways and cellular responses. The other category encompasses a range of functional associations that do not fall strictly under binding or modification. Gene co-expression patterns are also rated.

As seen in Figure 2, estrogen receptor (*ESR1*) encodes estrogen receptors and acts as a transcription factor. Chorionic gonadotropin, beta polypeptide 3, 5, and 8 (*CGB3*, *CGB5*, *CGB8*) are members of the glycoprotein family that encode subunits of chorionic gonadotropin. The chorionic gonadotropin alpha (*CGA*) chain gene is a human glycoprotein that encodes the alpha subunit. Histamine H2 receptor (*HRH2*) encodes for messenger molecule HRH2, which is released from mast cells and stimulates gastric acid secretion, important for digestion. Guanine nucleotide-binding protein, alpha-stimulating activity polypeptide (*GNAS*) is a complex gene locus and a key component of G protein-coupled receptor-regulated adenyl cyclase signal transduction pathways. The Lutropin subunit B (LHB) gene is a glycoprotein member that encodes the luteinizing hormone beta subunit. The glucagon (*GCG*) gene encodes a hormone that helps regulate glucose by triggering glucose production by the pancreas. Phosphodiesterase 4B (PDE4B) encodes proteins that regulate signal transduction. Cannabinoid receptors 1 and 2 (*CNR1* and *CNR2*) encode cannabinoid receptors. Cannabinoids are psychoactive ingredients in marijuana involved in mood and cognition, which also play a role in eating behavior. Sodium-dependent noradrenaline transporter (*SLC6A2*) encodes a member of the sodium neurotransmitter transporter family responsible for the reuptake of norepinephrine into presynaptic nerve terminals. Chymotrypsin-like elastase family member 3B (*CELA3B)* encodes proteases that hydrolyze elastin required for digestion. The protease serine 2 preprotein (*PRSS2*) gene encodes the protein that encodes trypsinogen required for digestion. Interleukin-4 (*IL-4*) encodes the cytokine produced by T cells, important for combating infections and plays a role in inflammation. Interferon-induced transmembrane protein 3 (*IFITM3*) encodes proteins in the family of interferon-induced antiviral-encoded RNA transcription and translation, and transport factor protein (*RTRAF)* encodes the protein that is involved in RNA binding. Myosin light polypeptide 6 (*MYL6*) encodes a motor protein in the hexametric ATPase of the myosin protein for smooth muscle development. Claudin-7 (*CLDN7*) encodes a protein in the claudin family that contributes to tight junction strands and serves as a physical barrier. Secretoglobin (*SCGB2A1*) encodes the protein involved in the androgen receptor signaling pathway. 5-Hydroxytryptamine receptor 1B (HTR1B) encodes the G protein receptor for serotonin, an important brain neurotransmitter. The HTR1B receptor manages the release of serotonin and dopamine with acetylcholine in the brain. The 26S proteasome non-ATPase regulatory subunit 8 (*PSMD8*) encodes a proteasome complex. Calcitonin *(CALCA*) encodes a related peptide as a hormone synthesized by the parafollicular cells of the thyroid. It causes a reduction in serum calcium, an effect opposite to that of the parathyroid hormone (PTH). For example, both GNAS and GCG are encoded proteins that play specific roles in hormone function and sensitivity, glucose production, and the regulation of obesity and eating behavior, as described in Table 1.

### 3.3. BioGRID Protein–Protein Interactions and Functions

The BioGRID (Biological General Repository for Interaction Datasets) program was used to identify and validate protein–protein and genetic interactions found in our other in silico studies using STRING and Pathway Commons programs and databases. The BioGRID program searches for protein–protein interactions and is used to search for and investigate GLP1R, as performed in the other web-based programs. We found that Gene Ontology biological processes identified included the activation of adenylate cyclase activity, cAMP-mediated signaling, energy reserve metabolic processes, the positive regulation of cytosolic calcium ion concentration, regulation of insulin secretion, and small molecule metabolic processes as the top significantly listed processes. For Gene Ontology molecular functions, we found transmembrane signaling receptor activity was listed, and plasma membrane for Gene Ontology cellular components.

## 4. Discussion

### 4.1. Background and Integrated Genetic and Protein Analysis of GLP1R

Genetic, molecular, and protein functions and interactions of *GLP1R* were studied using web-based programs and existing databases for the actions of semaglutide. STRING web-based protein–protein interactions, illustrated in Figure 1, showed 10 first-tier-associated protein nodes. The highest interaction was glucagon (GCG), having a likelihood score of 0.99 in the STRING dataset and followed by gastric inhibitory peptide (GIP). They have major roles in insulin secretion and control for glucose homeostasis when disturbed, leading directly to diabetes mellitus, appetite dysregulation, and obesity. Furthermore, adenylate cyclase activation and the modulation of G protein-coupled receptor signaling were most frequently utilized by proteins interacting with GLP1R agonists and increased intracellular cAMP acting as a secondary messenger, playing a role in insulin secretion. The most common molecular functions identified were hormones and peptide receptor binding activities playing a role in endocrine signaling cascades. Hypoglycemia was the most frequent disease–gene association found, which reflected the potent impact of insulin and other glucose-lowering agonists, such as semaglutide, in playing a role in treatment. These biological processes and functions must be normal for successful treatment with GLP1R agonists.

The second- and third-tier STRING analyses identified 20 extra associated protein nodes showing more complex protein–protein interactions with diverse involvement. These included key metabolic regulators GIP, GCG, and VIP, with additional functions for glycogenolysis, with the involvement of natriuretic peptide required for diuresis, natriuresis, and vasodilation. Adenylate cyclase-modulating G protein-coupled receptor signaling pathways are utilized by other associated proteins for related conditions such as hypoglycemia, thereby stimulating glucose utilization, energy expenditure, and weight loss as the most frequently observed disease–gene association.

The endocrine system was the most frequent disease–gene association found in the third-tier study. The adenylate cyclase-modulating G protein-coupled receptor signaling pathway was most utilized by the proteins identified in first- and second-tier testing. With the progression from 10 (first-tier) to 30 (third-tier) testing associated protein nodes, a clear pattern emerged, specifically the pathways affecting glucose regulation along with energy homeostasis. The involvement of the adenylate cyclase-signaling pathway across all three interactive tiers reinforces the central role of this signaling cascade and GLP1R function. This pathway appears crucial for promoting insulin secretion via increased cAMP levels as well as regulating appetite and satiety through the hypothalamic and central nervous system. With increased cAMP production, a decrease in feeding is seen over time, thus indicating cAMP’s significance as a second messenger for appetite through MC4R-expressing neurons in the paraventricular nucleus (PVH) neurons, which promote satiety [15,16,17,18,19,20].

Major interactors, including GNAS and GCG, were also identified in the Pathway Commons gene–gene interactions database, understandably supporting the role of glucose, insulin production, and metabolism as key factors. The shared pathways lend evidence that GLP1R activity impacts insulin secretion, glucose homeostasis, and signaling, and when disturbed, it leads to both type 2 diabetes and weight gain. The recurrence of these identified processes across the networks, as described, also supports the role of GLP1R agonists in achieving weight loss [21,22]. Hence, several findings were similar and reinforced the relevance of GLP1R agonists as described signaling pathways, functions, and processes for treating type 2 diabetes and also obesity.

Among the molecular functions of protein nodes, hormone activity showed the highest enrichment for 10 and 20 related nodes, while peptide hormone binding showed the highest enrichment when comparing all 30 protein nodes. Among the proteins involved in hormone activity, gastric inhibitory polypeptide (GIP) was notable. It has insulin-secretory effects, which inevitably lead to weight loss [21] and is utilized by GLP1R agonists. A related interactive gene (*GNAS*) is of interest, and its encoded protein was identified by both STRING and Pathway Commons web-based analyses. It plays critical roles in signaling associated with GLP1R function, particularly through its influence on intracellular cAMP levels involving multiple endocrine and other functions.

### 4.2. Related Obesity and Metabolic Genetic Factors with Disease Contributions

One of the most important obesity-related and metabolic genes recognized by our in silico analysis was *POMC* (pro-opiomelanocortin), a precursor synthesized in the corticotrophs of the anterior pituitary. The POMC protein is central in controlling appetite (satiety) [15,17]. This polypeptide is cleaved by prohormone convertases, which generate several peptides [17], including alpha-melanocyte-stimulating hormone (alpha-MSH), required for melanin production, appetite regulation at the satiety center, and sexual behavior, and adrenocorticotropic hormone (ACTH) produced by the pituitary that stimulates adrenal glands to release cortisol and beta-endorphin or opioid peptides. Alpha-MSH and melanocortin-4 receptor (MC4R) are related and involved in both pigment production and satiety, with the interaction of POMC for controlling appetite and playing a key role in the leptin-regulated melanocortin circuit essential for central energy regulation [18,19,20,21,22]. In connection with the inhibition of NPY/AgRP/GABA neurons, leptin exerts an anorexigenic effect with decreased eating behavior in humans. Conversely, a second neuroendocrine peptide known as ghrelin, a sole appetite-regulating peptide produced by the stomach, binds to the growth hormone receptor, which induces the activation of the AgRP-related neurons in the feeding center of the hypothalamus. This inhibits POMC neurons in the arcuate nucleus, thereby exerting an orexigenic or increased eating effect [15,16,17,18,19,20,21].

The *GNAS* gene encodes the stimulatory alpha subunit of the heterotrimeric G protein complex network of signaling pathways, thereby influencing many cellular functions that regulate hormone activities [15]. GNAS also showed both binding and co-expression properties. In addition, the G protein subunit encoded by GNAS or Gs alpha stimulates the activity of adenylate cyclase, leading to the production of cAMP from ATP [23]. It controls the production of several hormones with the regulation of endocrine glands such as the thyroid, pituitary, ovaries, testes, and adrenal glands. It is also critical in developing and regulating bone growth [22,23]. GNAS also interacts with pathways regulating parathyroid hormone activity and calcium homeostasis, muscle function, bone development, and stature.

The stimulation of GNAS further enhances cAMP signaling and is generally beneficial for insulin secretion and glycemic control. However, the excessive or prolonged activation of GNAS through chronic GLP1R agonist use could potentially disrupt calcium metabolism by altering parathyroid hormone responsiveness or signaling efficiency. Such dysregulation may contribute to imbalances in calcium and phosphate levels, affecting bone density, health, and neuromuscular function. Examples of GNAS-related disorders include Albright hereditary osteodystrophy and McCune–Albright syndrome, which affects bones and skin, and other conditions such as pseudo-hypoparathyroidism and pseudo-pseudo-hypoparathyroidism. These disorders often result in short stature, obesity, intellectual disability, abnormal craniofacial development, endocrine dysfunction and resistance, along with short metacarpals/metatarsals [15,24,25]. These disorders result from the decreased activity of the G protein complex Gs alpha subunit, causing hypocalcemia and hyperphosphatemia as noted in our analysis using the Pathway Commons web-based program. Further evidence supports the role of several hormones impacting growth and development; appetite regulation and metabolism; levels of insulin, glucagon, glucose, and calcium; and the involvement of other related peptides having multiple interactions described in our study.

The genetic analysis of GLP1R-associated proteins also highlights a significant influence on pancreatic function, blood glucose levels and regulation, and on proopiomelanocortin (POMC), which directly impacts satiety and energy levels [26]. Proteins such as gastric inhibitory polypeptide (GIP), glucagon (GCG), ADCYAP1, and its receptor (ADCYAP1R) were also shown to play direct roles in pancreatic beta-cell activity and insulin production. Through the cAMP signaling pathway, a recurrent molecular mechanism was identified across all STRING interaction networks. This signaling pathway enhances both insulin synthesis and release underlie the physiological basis for GLP1R agonist efficacy in lowering blood glucose. The consistent appearance of insulin secretion as a KEGG pathway with hormone activity in our study, as a recognized molecular function, affirms the central role of these genes in pancreatic endocrine regulation in weight management [27].

The combined results of our study with three web-based programs and databases suggest meaningful roles of GLP1R-associated genes and their encoded proteins in regulating intestinal motility, with vasoactive intestinal peptide (VIP), adenylate cyclase-activating polypeptide type 1 (ADCYAP1), and its receptor (ADCYAP1R) being notable, with their effects on smooth muscle relaxation in the gastrointestinal tract. VIP also promotes vasodilation and reduced smooth muscle tone, which would further contribute to slower intestinal transit. This action aligns with the observed function of GLP1R agonists in delaying gastric emptying, mechanisms that prolong satiety and support weight loss. Additionally, the adenylate cyclase-modulating G protein-coupled receptor signaling pathway supports increased intracellular cAMP levels. The association of these pathways and molecular functions with GLP1R and related proteins found in our study from multiple sources underscores the gastrointestinal effects of GLP1R agonists, with the ability to reduce motility and enhance nutrient absorption, influencing weight gain.

### 4.3. Related Mechanisms and Clinical Implications of GLP1R Agonists in Diabetes and Obesity

Sources from the literature and medical care activity demonstrate the common use of GLP1R agonists to treat type 2 diabetes and obesity. However, practical concerns have been raised about factors that could impact optimal therapeutic care, which would improve success and outcomes. For example, the disease(s), stage of disease process, and/or the cause or biology of the patient with the disease may negatively impact the functional mechanisms of the therapeutic drug, its effectiveness for both short- and long-term use, and potential comorbidities over time. These concepts were the driving motivation for our undertaking this in silico approach to study the integrated genetic and protein mechanisms of GLP1R and how the described findings or other factors could contribute to or impact patient care.

Both obesity and type 2 diabetes mellitus share genetic–pathological mechanisms with hundreds of genes implicated in playing a role [6,7,20,28,29,30,31]. Over 200 genome-wide studies and meta-analyses associated with type 2 diabetes and related disorders have shown dozens of genes and have been successfully replicated [6]. Other reports using genome-wide expression and integrated functional analysis of adipocytes collected from lean, obese, and type 2 diabetic subjects found a total of 1932 unique differentially expressed genes across type 2 diabetic and obese subgroups. The enriched terms were in response to lipids, the regulation of cell death and differentiation, insulin, phosphorylation, fatty acid transport, glutamate receptor binding, and the response to hormones and glucose [31,32,33]. These findings were similar to our in silico analysis results when studying the GLP1R gene and its encoded protein. We also found additional genes in our study, such as GNAS, which plays a role in hormone regulation, response, and sensitivity, as well as other genes coding for neuropeptides, transmitters, and receptors impacting brain function and behavior (e.g., serotonin, dopamine, cannabinoid receptors) and infection or inflammatory markers (e.g., cytokines, interferon).

Insulin resistance in obese individuals may be due to elevated levels of non-esterified fatty acids, cytokines, hormones, and inflammatory involvement [33,34,35,36]. Fatty acids secreted from adipose tissue and obese individuals may be key when connecting impaired pancreatic beta cell function, insulin resistance, and inflammation in type 2 diabetes with comorbidities and disease progression [31,33,34,35,36]. Furthermore, genes such as *PPARG*, *UCP3*, *ENPP1*, *POMC,* and *FTO* are consistently identified as obesity-related genes [20,29,31,37,38,39] with key molecular mechanisms. If these and hundreds of other recognized obesity- and/or diabetic-related genes are altered, then they may lead to a disease-state not responding to conventional drug treatment or may have ‘side-effects’. Ideally, understanding the genetic and pathophysiology will further contribute to developing strategies and surveillance to treat to better treat the patient and their needs.

### 4.4. GLP1R-Related Studies in Mice

Laboratory observations of the brain and other organs or tissues in mice suggest that long-term diabetes and obesity-related pathophysiologic states alter neurotransmitter uptake and transmission, including neuronal excitatory/inhibition biological processes and pathways related to oxidative stress and inflammation. The same mechanisms and reported pathways in neuroprotection, synaptic plasticity, and energy metabolism in genetically diabetic mice were found, which may require further attention when treating both type 2 diabetes and obesity in humans using GLP1R agonists, specifically with the extended use of the drug and medical care. Several protein interactions and functions reported in mice involved the regulation of calcium and other cation binding, insulin, glucose, and metabolic changes, growth factors and control, inflammation, cellular growth, and the regulation of hormone signaling pathways. We also identified similar findings using an in silico or web-based approach when analyzing GLP1R in humans, further supporting the role of this gene, its encoded protein, and the interactions in diabetes and obesity in both humans and mice. More experimental research is needed to include fresh tissue and organs for study in humans, with follow-up and validation of evidence reported in animal models, but challenging or difficult to undertake in humans.

### 4.5. Study Limitations, Perspectives, and Future Studies

Study limitations are recognized, including the central use of existing information from the literature stored in databases and analyzed with web-based programs accessed by searching web-based databases and in silico approaches. This analysis was undertaken without the use of GLP1R proteomics-based laboratory studies or experiments, which were considered outside our scope of work. We searched web-based programs for gene–gene and protein–protein interactive databases in humans only. Our goal was to examine the integrated genetic and protein mechanisms underlying GLPR agonist treatment and outcomes.

We identified and analyzed the genetic–functional mechanisms and protein interactions described in our study of the GLP1R gene and protein with disease associations and the treatment of type 2 diabetes and obesity. We then raise concerns about how functional mechanisms may be affected by disease causation and progression. The patient’s disease and level of involvement may impact the success of treatment with a GLP1R agonist, the other medications used, and disrupt their biology, which may be present depending on the patient. We also discuss the importance of dual-GLP1R-agonist treatment by adding a second drug for treating other disease mechanisms related to disease progression and long-term care. The patient’s disease state, drug response, and resistance, along with existing or new comorbidities, could be monitored on a regular basis. More research is needed to identify the downstream effects or consequences of drug therapy alone or in combination with other drugs. The genomics and proteomics data from collected tissues at baseline and studied at different stages of disease progression or treatment from affected and control subjects would be informative. Recording and comparing the clinical and biological findings over time with potential side-effects or other expected or unexpected health changes with age would be important information in the use of GLP1R agonist(s) for treating type 2 diabetes and obesity in humans, now and in the future.

Studies on the extended use of GLP1R agonists in treating humans and their diseases should include drug resistance or efficacy measures along with screening for causative factors, including genetics, preferably prior to treatment and surveillance. The establishment of optimal treatment plans with GLP1R agonists and dual options will require surveillance for long-term care. In addition, monitoring existing or new comorbidities and clinical findings would be recommended for standard care related to our web-based gene and protein and literature review findings with GLP1R. Our in silico analysis of the GLP1R gene and its encoded protein identified interactions involving other related genes and proteins; these interactions, if disturbed, may result in hormone imbalances, digestive problems, inflammation, electrolyte or neurotransmitter disturbances, behavior and cognitive issues, including decline or dementia, retinal, cardiac, and kidney issues, or bone, muscle, and energy-metabolic changes. These findings are also noted in type 2 diabetes and/or obesity, indicating an inter-relationship between the two disorders, based on a common genetic–physiological background. When this background is disturbed, then these disorders arise, obviously involving GLP1R activity, as evidence which shows treatment success with GLP1R agonists. The obvious task is testing the genetics and biology of affected patients to discover the causative factor(s) contributing to the decreased function or mechanism undertaken by GLP1R, which leads to these related conditions.

There is growing data in genetically induced diabetic mice that have shown that treatment with GLP1R equivalent-related drugs can lessen the characteristic disease findings. Detailed genetic testing options do exist, are readily available in the clinical setting, and should be recommended preferably prior to drug administration. If a GLP1R gene defect is present or found in one of the other related genes involved in the mechanisms of action, then the likelihood of success of a single drug would be diminished. Conversely, a similar outcome may occur if the patient has a defect of a gene playing a role in obesity (e.g., FTO, POMC, etc.). Alterations in the patient’s medical history, course, or progression of disease prior to treatment and previous clinical treatment plans and outcomes should be carefully recorded and monitored for longevity with the documentation of success or lack thereof with such treatment to generate more research and useful data for treating diabetes and obesity.

Hundreds of recognized genes and their encoded proteins with inter-patient diversity may play a role in the causation and progression of both type 2 diabetes and obesity. Genetic testing could be addressed with whole exome/genomic sequencing analysis or disease-specific gene panels. The results could impact treatment selection and success. This genomic study would also examine variants for their encoded proteins and related functions that impact GLP1R agonist treatment. This approach could lead to a more comprehensive, individualized care and treatment plan for medical management and disease surveillance in the future. If not currently available, we would encourage treatment registries for the recording of natural history data, evidence of drug resistance, status of existing or new disease comorbidities, and quality-of-life measures when treating type 2 diabetes mellitus and/or obesity with GLP1R agonists, as unexpected clinical observations have been noted previously that may contribute to future success. Hence, therapeutic agents should be monitored for continued success, recording unforeseen factors that could alter continued optimal care. In addition, pharmacogenetics testing would be encouraged for the selection and management of medication use. Disease mechanisms or actions could be affected by disease progression prior to treatment or during treatment.

### 4.6. Summary

The GLP1R gene and its associated protein network orchestrate complex signaling mechanisms as described in our in silico studies that improve glycemic control and promote weight loss through metabolic, endocrine, gastrointestinal functional pathways, and related factors. We utilized three separate, searchable, and established web-based programs and databases to analyze the GLP1R gene and its protein–protein interactions, molecular functions, biological processes, pathways, cellular components, and associated diseases. Our study identified and described current factors and mechanisms of action and processes for commonly used GLP1R agonists in treating type 2 diabetes mellitus and obesity.

Adenylate cyclase-modulating G protein-coupled receptor signaling pathways were often identified across all three STRING tier-level analyses and supported by two separate searchable gene and protein web-based programs for characterization of GLP1R agonist activity. Other functions were attributed to genes and/or proteins, including GIP, GNAS, GCG, SCT, POMC, VIP, GLP1, and ADCYAP1R, with some identified for the first time (e.g., GNAS, CALCA, CALCB, HRH2, CNR1, CNR2) for hormone control and sensitivity, electrolyte balance, and neurological/behavioral function. Their actions include glucose regulation, insulin production and secretion, hypothalamic function, satiety control via gastrointestinal motility, electrolytes, and related peptides and hormones. Organs and tissues could be affected, including the skeletal system, sex and other endocrine glands, and neuropeptides and brain function, along with an impact on muscle mass and strength. These factors would relate directly to GLP1R agonist treatment.

The recognition of molecular functions and protein insights will enhance the understanding of therapeutic mechanisms and the impact on clinical manifestations, particularly in the use of GLP1R agonists. These genetically related findings, when examined, would impact clinical outcomes driven by genetic heterogeneity and diversity in our population. In addition, we highlighted the potential for unforeseen systemic effects with long-term use and the risk for resistance, particularly within the endocrine and metabolic systems, calcium and electrolyte regulation and function, related gut peptide and hormone imbalances, brain, cardiac, behavioral, and neuromuscular issues, muscle mass and strength, gastrointestinal motility and function, as well as vision and kidney issues. In addition, bone structure, density, and health should be further monitored with long-term follow-up, clinical evaluations, and outcome measures.

The limitations of our in silico analyses were also noted, and future studies are proposed to raise questions that may impact GL1PR agonist use, both short- and long-term, and its related functional mechanisms. We would encourage the use of treatment registries for recording natural histories, drug response, other medications used, and existing or new comorbidities. The enriched functional molecular mechanisms and protein interactions of GLP1R were identified and described a positive impact with agonist use, indicating success in treating type 2 diabetes and obesity. Other factors were described as contributing to success or failure. Therapeutic agents and medical care should be continuously monitored for success in order to recognize and address unforeseen factors that could alter optimal care.

## Figures and Tables

**Figure 1 cimb-47-01007-f001:**
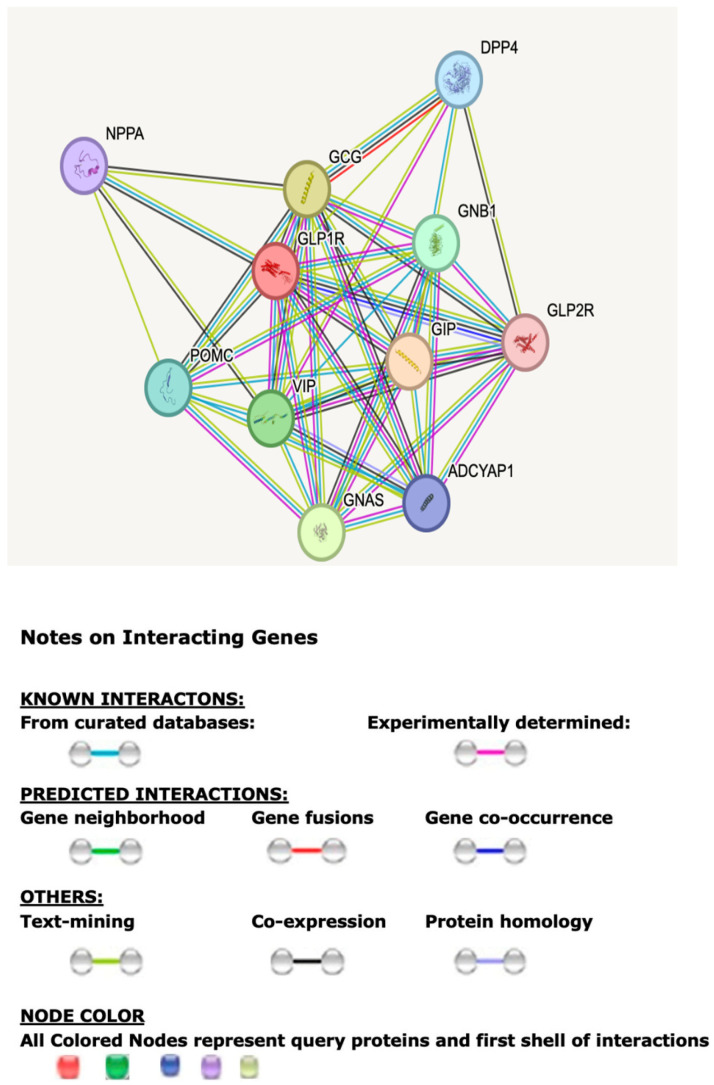
STRING protein–protein interaction with first-tier network for the GLP1R gene and functional interactions involving 10 associated protein nodes and 44 edges with their predicted functional interactions, including shared biological processes, pathways, and molecular functions (https://string-db.org). Network nodes represent proteins with splice isoforms or post-translational modifications collapsed into each node for all proteins produced by a single protein-coding gene. Edges represent protein–protein associations that are considered specific and meaningful, or proteins jointly contributing to a shared function.

**Figure 2 cimb-47-01007-f002:**
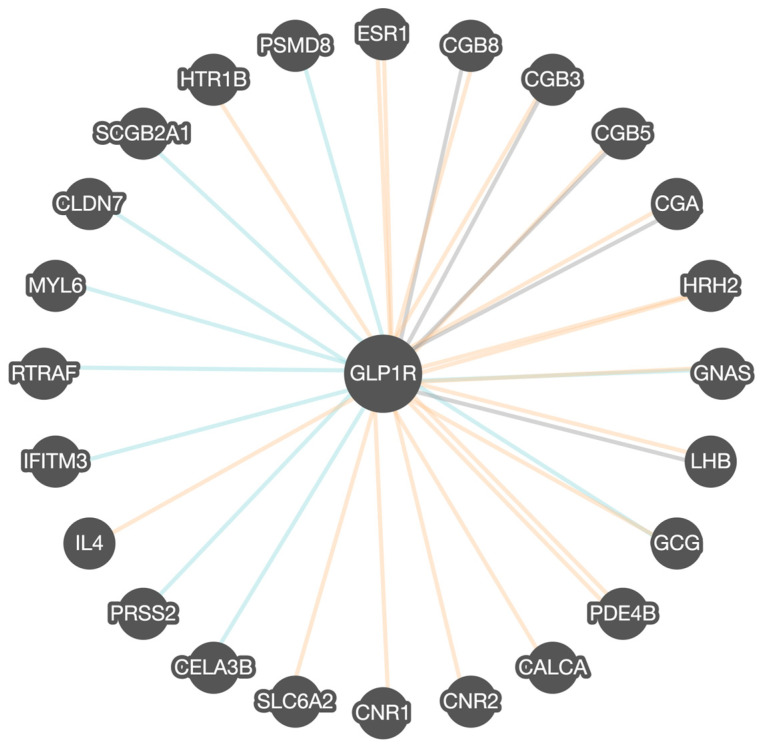
Pathway Commons GLP1R gene–gene functional interactions determined via binding (blue lines), co-expression (orange lines), and other interactions (gray lines). Multiple orange lines for *ESR1* and *PDE4B* genes represent multiple co-expression patterns. The *GLP1R* gene is highly related to G-protein-coupled receptor signaling and activation of adenylate cyclase. This involves increased intracellular cAMP. Of the 24 genes associated with *GLP1R*, 15 showed binding ability, 16 showed modification or co-expression patterns, and 7 showed other interactions. (https://apps.pathwaycommons.org/interactions/?source=GLP1R). Accessed on 28 July 2025.

**Table 1 cimb-47-01007-t001:** Protein symbols and description of the top ten significantly associated proteins in the first-tier STRING analysis for GLP1R *.

Protein Symbol	Description
**GIP**	Gastric inhibitory polypeptide is a stimulator of insulin secretion and a poor inhibitor of gastric acid secretion.
**GCG**	Glucagon is involved in glucose metabolism and homeostasis, with regulation of blood glucose by increasing gluconeogenesis and decreasing glycolysis, and plays an important role in appetite control. Glucagon is a counterregulatory hormone of insulin and raises plasma glucose levels in response to insulin-induced hypoglycemia.
**GNAS**	Guanine nucleotide-binding protein G(s) subunit alpha isoforms XLas or G protein function as transducers in the signaling pathways controlled by G protein-coupled receptors. Signaling involves the activation of adenylyl cyclase, which increases the levels of the signaling molecule cAMP and plays a role in several obesity-related genetic disorders with hormone disturbances.
**VIP**	Vasoactive intestinal peptide causes vasodilation, lowers blood pressure, stimulates myocardial contractility, increases glycogenolysis, and relaxes smooth muscle of the trachea, stomach, and gallbladder.
**GNB1**	Guanine nucleotide-binding proteins (G proteins) serve as modulators or transducers in various transmembrane signaling systems. The beta and gamma chains are required for the GTPase activity.
**POMC**	Proopiomelanocortin stimulates the adrenal glands to release cortisol and generates melanocyte-stimulating hormone beta that increases skin pigmentation by increasing melanin production in melanocytes. POMC is an important obesity-related hormone polypeptide impacting satiety in the hypothalamus.
**DPP4**	Dipeptidyl peptidase 4 membrane form is a cell surface glycoprotein receptor involved in the costimulatory signal essential for T-cell-receptor-mediated T-cell activation.
**ADCYAP1**	Pituitary adenylate cyclase-activating polypeptide 27 binds to its receptor, which causes the activation of G proteins and stimulates adenylate cyclase in pituitary cells. It promotes neuron projection development through the RAPGEF2/Rap1/B-Raf/ERK pathway. In chromaffin cells, it induces an increase in intracellular calcium concentrations and neuroendocrine secretion. It induces insulin secretion in pancreatic beta cells.
**NPPA**	Natriuretic peptide precursor A is a hormone significant in cardiovascular homeostasis through the regulation of natriuresis, diuresis, and vasodilation.
**GLP2R**	Glucagon-like peptide 2 receptor and activity mediated by G protein and adenylyl cyclase.

* STRING website (www.string-db.org).

**Table 2 cimb-47-01007-t002:** STRING: predicted functions for GLP1R with first-tier analysis of ten associated protein nodes *.

Biological Process (Gene Ontology)	^A^ CIN	^B^ Strength	^C^ Signal	^D^ FDR
Adenylate cyclase-activating G protein-coupled receptor signaling pathway	7 of 145	1.94	3.57	3.56 × 10^−9^
Adenylate cyclase-modulating G protein-coupled receptor signaling pathway	8 of 232	1.79	3.39	1.08 × 10^−9^
Regulation of hormone levels	7 of 525	1.38	1.64	1.18 × 10^−5^
G protein-coupled receptor signaling pathway	10 of 1174	1.18	1.6	3.21 × 10^−8^
Activation of adenylate cyclase activity	3 of 31	2.24	1.53	0.0013
**Molecular Function**	**CIN**	**Strength**	**Signal**	**FDR**
Glucagon receptor activity	2 of 3	3.08	1.66	0.00100
**KEGG Pathway**	**CIN**	**Strength**	**Signal**	**FDR**
cAMP signaling pathway	8 of 207	1.84	4.04	9.48 × 10^−12^
Insulin secretion	5 of 82	2.04	3.27	7.56 × 10^−8^
Neuroactive ligand-receptor interaction	7 of 329	1.58	2.58	2.06 × 10^−8^
**Reactome Pathway**	**CIN**	**Strength**	**Signal**	**FDR**
Glucagon-type ligand receptors	8 of 33	2.64	8.05	6.35 × 10^−17^
G alpha(s) signaling events	9 of 157	2.01	5.39	1.06 × 10^−14^
GPCR ligand bonding	9 of 459	1.55	2.91	6.69 × 10^−11^
Glucagon-like peptide-1 (GLP1) regulates insulin secretion	4 of 42	2.23	2.85	2.15 × 10^−6^
**Disease-gene Association**	**CIN**	**Strength**	**Signal**	**FDR**
Hypoglycemia	4 of 22	2.51	2.88	3.58 × 10^−6^
Hyperglycemia	3 of 14	2.58	2.0	0.00020
Type 2 diabetes mellitus	3 of 25	2.33	1.7	0.00064
Carbohydrate metabolic disorder	4 of 268	1.43	1.0	0.0050
Disease of metabolism	6 of 1076	1.0	0.73	0.0050

* STRING website (**www.string-db.org**). **^A^ CIN** (Count in network) indicates how many proteins in the network are annotated with a particular term and how many proteins in total (in network and in the background) have this term assigned to this variable, per category (Biological Process, Molecular Function, etc.). **^B^ Strength**(Log10 (observed/expected)). This measure describes how large the enrichment effect is within the ratio between (i) the number of proteins in the network that are annotated with a term and (ii) the number of proteins expected to be annotated with this term in a random network of the same size. **^C^ Signal** is defined as a weighted harmonic mean between the observed/expected ratio and -log (FDR). FDR, or false discovery rate, tends to emphasize larger terms, due to their potential for achieving lower *p*-values, while the observed/expected ratio highlights smaller terms, which have a high foreground to background ratio but cannot achieve low FDR values due to their size. **^D^ FDR** is a statistical measure that examines the significance of enrichment. Shown are ***p*****-values** corrected for multiple testing within each category. All FDR-derived values in Table 2 are significant at *p* < 0.05.

## Data Availability

The original contributions presented in this study are included in the article. Further inquiries can be directed to the corresponding author.

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
