# Peer review of "Integrated Genetic and Protein Mechanisms Underlying Glucagon-like Peptide-1 Receptor Agonists in Treating Diabetes Mellitus and Weight Loss"

_cimb, 2025, doi:10.3390/cimb47121007_

Round 1
Reviewer 1 Report
Comments and Suggestions for Authors
Please see the attached word document. Thank you.

Author Response
November 17, 2025
Current Issues in Molecular Biology
Dear Editors and Reviewers,
We appreciate the time and thoughtful review of our Perspective manuscript, titled “Integrated Genetic and Protein Mechanisms Underlying Glucagon-Like Peptide-1 Receptor Agonists Using In Silico Analysis” submitted to Current Issues in Molecular Biology. We appreciate the detailed feedback and suggestions for improving and revising our manuscript. We have carefully addressed each comment using a point-by-point response noted in red and italics, removed redundant figures, repetitive text, extraneous information and references:
Reviewer 1: Comment 1- The introduction could be strengthened by first outlining the global burden of type 2 diabetes mellitus and related metabolic complications, then introducing GLP1R agonists as a therapeutic strategy for and diabetes and weight loss. This would provide a clearer big-picture context for readers before presenting the molecular mechanisms in detail.
Response: Thank you for the suggestions. We substantially revised and shortened the entire manuscript beginning with the Introduction section by providing a clearer global perspective on the burden of type 2 diabetes mellitus and its associated metabolic complications with obesity. After establishing this clinical context, we introduced GLP1R agonists as a therapeutic strategy for glycemic control and weight reduction. This restructuring now positions the molecular discussion in a broader clinical framework and improves the narrative flow for readers less familiar with diabetes epidemiology.
Reviewer 1: Comment 2- In the Methods section, I suggest dividing the content into smaller section parts based on each database or analytical tool used (for example, STRING, Pathway Commons, OMIM, and GeneCards). This would make the section easier to follow and improve clarity and reproducibility.
Response: We reorganized the entire section by creating individual subsections as suggested corresponding to each web-based analytic tool and database used, including STRING, Pathway Commons and BioGRID along with literature sources. Each subsection now clearly describes access dates, versions and search parameters, relevant filters, and data extracted. This restructuring and text revision provides a more systematic workflow presentation which enables readers to follow our analytic approach and data descriptions.
Reviewer 1: Comment 3- In Section 3.1 (STRING Protein-Protein Interactions, Functions and Analysis), I suggest integrating the legend for interaction types and node colors directly into Figure 1 or placing it more closely below the network. This would make the figure more self-explanatory and easier to interpret without referring back to the text.
Response: We enlarged the node color scheme and interaction-type legend and placed under Figure 1 as requested. We removed the old Figures 2 and 3 to avoid any redundancy and to shorten the length of the manuscript. Figure 1 now includes a revised format for clarity and reader usability.
Reviewer 1: Comment 4- In Figure 4, certain nodes (e.g., ESR1, PDE4B) show duplicate orange edges. If these represent multiple co-expression records or distinct datasets, please clarify this in the legend or text. Otherwise, the duplication may confuse readers.
Response: We have eliminated old Figures 2 and 3. The original Figure 4 now is renumbered as Figure 2. The duplicate edges in orange represent multiple co-expression relationships identified across independent gene-gene interactive datasets within Pathway Commons. The legend has been updated to explicitly explain this, and the caption now includes a clarification describing why some genes exhibit multiple orange lines.
Reviewer 1: Comment 5- In Figure 4, the legend should include the definition of the gray lines representing the ‘other interactions,’ as their meaning is unclear
Response: We added further clarification to the old Figure 4 now renumbered as Figure 2 to both the legend and manuscript text to define “other interactions.” These represent functional relationships recorded in Pathway Commons that are neither direct binding nor co-expression interactions, such as regulatory or contextual functional associations. This updated explanation ensures that all line colors within the new Figure 2 are fully interpretable.
Reviewer 1: Comment 6- In Section 3.2 and Figure 4, the text lists GNAS among the interacting genes but does not define or describe it and also in the description of GCG, the abbreviation is placed in parentheses after the full name for most genes, but here it appears reversed (‘GCG (glucagon)’). For consistency with the rest of the text, it should read ‘glucagon (GCG)’.
Response: We added a concise and more informative description of GNAS, including its molecular function, signaling relevance, and importance to GLP1R-related cAMP pathways. We also corrected the formatting of all gene names to maintain consistency with genomic and protein nomenclature standards, including changing “GCG (glucagon)” to “glucagon (GCG)” for alignment with the remainder of the manuscript.
Reviewer 1: Comment 7- In Section 3.2 and Figure 4, you list 24 GLP1R-associated genes, but only about 22 are explained. GNAS and GCG are mentioned without definition. It would be better to add a short description for these two or clearly note that they’re summarized in Table 1 to keep it consistent.
Response: Thank you for identifying the omission of two genes (GNAS and GCG) in the gene Results description paragraph. We have now included short summaries for both GNAS and GCG within the Results section to complete the explanations for all genes referenced. These additions ensure that readers have access to the description of each gene and its biological role within the GLP1R network. Cross references to Table 1 were also added as well.
Reviewer 1: Comment 8- Ensure consistent formatting: use italic caps for human genes in text (e.g., GNAS, GCG) and non-italic caps for proteins (GNAS, GCG).
Response: We performed a comprehensive formatting review of the manuscript to ensure consistent use of italicized uppercase symbols for human gene names and standard, non-italicized uppercase symbols for proteins. This correction has been applied throughout the text, tables, and figure captions. We also have standardized the formatting of titles and subheadings throughout.
Reviewer 1: Comment 9- In the Results section (preceding Table 1), the sentence ‘The GLP1R encoded protein is a glucagon-like peptide-1 receptor which is G protein coupled for glucagon-like peptide-1 (GLP1)’ is redundant and does not contribute new information. I suggest removing or rephrasing it, as this definition is already well established and explained earlier in the introduction.
Response: We removed the redundant definition of GLP1R from the Results section. This information was discussed in the Introduction and other sections and did not contribute new insights at this stage of the manuscript.
Reviewer 1: Comment 10- In Result, Section 3.1 the sentence( GLP 1 receptor is receptor for GLP1) it’s obvious redundant.
Response: We removed this phrase to improve conciseness and avoid redundancy. The revised section now focuses solely on new functional or analytical findings derived from our STRING analysis.
Reviewer 2 Report
Comments and Suggestions for Authors
Comments for Authors: A thorough bioinformatics investigation of GLP-1 receptor (GLP1R) agonists is presented in the study, utilising many public datasets (STRING, Pathway Commons, and BioGRID). The writers put a lot of work into compiling the data, and the topic is current. The current edition, however, lacks the critical depth and methodological clarity anticipated in Current Issues in Molecular Biology and is descriptive rather than analytical.
Important points that need to be revised:
- Focus and novelty: Describe the precise research gap that this work fills and emphasise the novel biological discoveries that go beyond the analysis of the GLP1R network.
- Techniques: Indicate the date of access, score criteria, database versions, and query parameters. Describe the steps taken to handle duplicate or low-confidence interactions.
- Analytical explanation: Provide biological interpretation (e.g., how specific nodes modify known GLP1R signalling cascades) rather than a list of network outputs.
- Discussion: Simplify to highlight important mechanistic discoveries. Cut down on repetition and speculative clinical extrapolations.
- Tables and figures: Boost layout, label readability, and resolution. Think about adopting simplified pathway diagrams or combining several STRING network levels.
- Formatting and references: Make sure they follow MDPI guidelines and fix any broken URLs.
|
Evaluation Criteria |
Response |
|
Does the introduction provide sufficient background and include all relevant references? |
Can be improved – Background is adequate but the rationale and study objectives should be more sharply defined. Current text is overly narrative and could be condensed. |
|
Is the research design appropriate? |
Must be improved – The study relies exclusively on in-silico database integration without adequate validation or comparative verification. The design needs stronger justification of novelty and analytical depth. |
|
Are the methods adequately described? |
Must be improved – The authors should clearly state database versions, inclusion/exclusion criteria, confidence thresholds, and data filtering methods to enhance reproducibility. |
|
Are the results clearly presented? |
Can be improved – Results are comprehensive but mainly descriptive. Figures and tables should be simplified and supported with analytical interpretation rather than restating database outputs. |
|
Are the conclusions supported by the results? |
Must be improved – The conclusions extrapolate beyond the scope of the computational data. The discussion must differentiate between data-based findings and theoretical implications. |
|
Are all figures and tables clear and well-presented? |
Can be improved – Figures are informative but overcrowded, especially STRING networks. Increase image resolution (≥300 dpi), enlarge labels, and refine legends for clarity. |
|
Quality of English Language |
The English is fine and does not require major improvement. Minor stylistic editing for conciseness and reduced redundancy is suggested. |
The manuscript could contribute much with these changes. However, it has to be significantly revised at this time before being reconsidered.
Author Response
November 17, 2025
Current Issues in Molecular Biology
Dear Editors and Reviewers,
We appreciate the time and thoughtful review of our Perspective manuscript, titled “Integrated Genetic and Protein Mechanisms Underlying Glucagon-Like Peptide-1 Receptor Agonists Using In Silico Analysis” submitted to Current Issues in Molecular Biology. We appreciate the detailed feedback and suggestions for improving and revising our manuscript. We have carefully addressed each comment using a point-by-point response noted in red and italics, removed redundant figures, repetitive text, extraneous information and references:
- Reviewer 2: Comment 1- Focus and novelty: Describe the precise research gap that this work fills and emphasise the novel biological discoveries that go beyond the analysis of the GLP1R network.
Response: We revised the manuscript throughout for brevity and consistencies including the Introduction and the Discussion sections. We more explicitly articulated the research gap our study addresses, specifically using updated functional interactions and molecular mechanisms related to highlighted GLP1R agonists widely used clinically and contributing factors that impact causation, treatment and medical care of patients with type 2 diabetes and obesity. We used established updated web-based programs and databases to search comprehensive integrated genetic and protein interactions using three separate programs (STRING, Pathway Commons and BioGRID) along with literature searches for GLP1R. This current comprehensive approach to search gene-gene and protein-protein interactions, biological processes, molecular functions, pathways, cellular components and gene-disease associations at one time have been lacking. We emphasize this novel contribution in identifying multi-tiered GLP1R-associated networks, newly recognized functional relationships, and less-characterized interacting genes such as CALCA, CALCB, CNR1, CNR2, and GNAS-related endocrine pathways playing a role in GLP1R activity and targeted by GLP1R agonists. The Discussion section has been revised and shorted with more clarity. Our intent was to identify potential new molecular mechanisms and newly enriched functions contributing to GLP1R and mechanisms of action for treatment agonists. We raised potential issues that may complicate therapy beyond what is found in the existing literature and discuss genetic factors and testing which may help in treating type 2 diabetes and obesity as well as response to treatment in which GLP1R is targeted by GLP1R agonists.
- Reviewer 2: Comment 2- Techniques: Indicate the date of access, score criteria, database versions, and query parameters. Describe the steps taken to handle duplicate or low-confidence interactions.
Response: An expanded Methods section was developed to further address these questions or concerns. The techniques and methodology with statistical analysis approach were further expanded to address several of these concerns raised when querying the updated web-based programs and databases in our study. The embedded statistical computational programs analyze the searched data and generate p values adjusted per category based on duplication data and confidence intervals as listed in our report in the Methods and Results sections. We now list the date of access (2025) and database versions (version 12 and 12.5-STRING). The statistical approach was expanded and literature cited. There have been at least 120 peer-reviewed publications when searching PUBMED for STRING-db indicating its establishment in the scientific community.
- Reviewer 2: Comment 3- Analytical explanation: Provide biological interpretation (e.g., how specific nodes modify known GLP1R signalling cascades) rather than a list of network outputs.
Response: We expanded the Discussion section to provide more useful data and interpretation that could impact clinical outcomes and measures with success of GLP1R agonist treatment and networks identified. We also raised concerns and how the concerns could be addressed when treating type 2 diabetes and obesity with GLP1R agonists. For each major interacting node (e.g., GNAS, GIP, VIP, POMC, CALCA/CALCB), we now describe known or plausible roles in GLP1R signaling, metabolic regulation, appetite control, insulin secretion, endocrine pathways, and potential clinical implications. We also added explanations for how specific genes modulate GLP1R signaling cascades, including cAMP-mediated activity, GPCR cross-talk, and neuroendocrine regulation.
- Reviewer 2: Comment 4- Discussion: Simplify to highlight important mechanistic discoveries. Cut down on repetition and speculative clinical extrapolations.
Response: Thank you for the suggestion. We streamlined the Discussion section by removing repetitive explanations of database methods and consolidating overlapping biological concepts. We reorganized the section to center on the most meaningful mechanistic discoveries: the dominant role of adenylate cyclase–modulated GPCR signaling across all analysis testing tiers, central nodes that regulate insulin secretion and appetite, endocrine and metabolic implications of novel interactors, relevance to type 2 diabetes and obesity pathophysiology. Speculative clinical statements were reduced or reframed more cautiously based on current evidence.
- Reviewer 2: Comment 5- Tables and figures: Boost layout, label readability, and resolution. Think about adopting simplified pathway diagrams or combining several STRING network levels.
Response: We eliminated figures to save space and improved figure clarity with improved layout throughout the manuscript. Specifically:
- Figures 2 and 3 were removed to eliminate redundancy and improve visual flow.
- Legends were revised for readability and completeness.
- Font sizes and color contrasts were increased for clarity.
- These updates enhance interpretability and ensure compliance to improve presentation with publication standards including developing subheadings per section for better accessing and readability.
- Reviewer 2: Comment 6- Formatting and references: Make sure they follow MDPI guidelines and fix any broken URLs.
Response: We completed a line-by-line formatting audit to ensure full compliance with MDPI standards. This included reference style corrections, hyperlink validation for all URLs, proper in-text citation formatting, table numbering, and figure placement. Any broken links were replaced with correct, active URLs. The cited references were checked for accuracy and citations.
We have also carefully proofread the manuscript for grammar and clarity and ensured that all figures, tables, and references adhere to the journal’s guidelines.
Thank you again for the opportunity to revise our work. We hope the revised manuscript meets the journal’s expectations and is now suitable for publication. Please do not hesitate to contact us if any further clarifications are needed.
Warm regards,
Lucas Francis B.S. (2nd year Medical Student) and Merlin G. Butler M.D. , PH.D.
lucas.francis@kansascity.edu and mbutler4@kumc.edu
Round 2
Reviewer 2 Report
Comments and Suggestions for Authors
I accept author response.